# Immunogenicity and Therapeutic Efficacy of a Sendai-Virus-Vectored HSV-2 Vaccine in Mouse and Guinea Pig Models

**DOI:** 10.3390/vaccines11121752

**Published:** 2023-11-24

**Authors:** Xiuxiu Ren, Wenhao Su, Shishi Li, Tingting Zhao, Qiufang Huang, Yinan Wang, Xiaojie Wang, Xiaohuan Zhang, Jiangbo Wei

**Affiliations:** Weijiangbo Laboratory, National Vaccine and Serum Institute, Beijing 101111, China; renxiuxiu@sinopharm.com (X.R.); suwenhao@sinopharm.com (W.S.); lishishi@sinopharm.com (S.L.); zhaotingting10@sinopharm.com (T.Z.); huangqiufang3@sinopharm.com (Q.H.); wangyinan1@sinopharm.com (Y.W.); wangxiaojie18@sinopharm.com (X.W.); zhangxiaohuan@sinopharm.com (X.Z.)

**Keywords:** herpes simplex virus type 2, Sendai virus vector, therapeutic vaccine, animal models

## Abstract

Background: To date, there is no licensed vaccine for preventing herpes simplex virus type 2 (HSV-2). The current treatment to address the infection and prevent its transmission is not always satisfactory. Methods: We constructed two recombinant vectors, one encoding HSV-2 glycoprotein D (gD, SeV-dF/HSV-2-gD) and one encoding HSV-2-infected cell protein 27 (ICP27, SeV-dF/HSV-2-ICP27), based on a replication-defective Sendai virus through reverse genetics, collectively comprising a combinatorial HSV-2 therapeutic vaccine candidate. The immunogenicity and proper immunization procedure for this vaccine were explored in a murine model. The therapeutic effect that helps prevent recurrent HSV-2 disease was evaluated in HSV-2-infected guinea pigs. Results: Both a robust humoral immune response and a cellular immune response, characterized by the neutralizing antibody titer and the IFN-γ level, respectively, were elicited in BALB/c mice. A further study of cellular immunogenicity in mice revealed that T lymphocytes were successfully enhanced with the desirable secretion of several cytokines. In HSV-2-seropositive guinea pigs, vaccination could reduce the severity of HSV-2 in terms of recurrent lesions, duration of recurrent outbreak, and frequency of recurrence by 58.66%, 45.34%, and 45.09%, respectively, while viral shedding was also significantly inhibited in the vaccine-treated group compared to the group treated with phosphate-buffered saline. Conclusions: The replication-defective recombinant Sendai viruses conveying HSV-2-gD and ICP27 proteins showed great immunogenicity and potential for preventing recurrent HSV-2 disease.

## 1. Introduction

Herpes simplex virus type 2 (HSV-2) is a major causal agent of human sexually transmitted infections, with a typical pattern of initial infection–latency–recurrence, of which, the recurrence can be lifelong with sporadic outbreaks. During the initial episode, HSV-2 may lead to genital herpes, ulcers, and lesions at the site of infection as well as several symptoms, including tingling, burning, and urination problems. After a series of activities at the genital skin, HSV-2 moves into the peripheral sensory nerve and settles down at the root ganglia through the nerve axons without destroying any neurons. The latency of HSV-2 is thereby established and maintained at the root ganglia in the form of nucleocapsids, which enables HSV-2 to avoid being recognized and eradicated by the host immune system [1]. Latent HSV-2 becomes reactivated via a hitherto uncertain mechanism and may cause recurrent episodes. At this time, HSV-2 travels back to the genital surface, and intermittent viral shedding may occur [2]. Both initial and recurrent infections can be subclinical, but the viral shedding is independent of the symptoms. Also, HSV-2 may enter the central nervous system in a retrograde manner, causing cephalitis, meningitis, and several neurological symptoms [3]. Neonatal HSV-2 infection is another problem resulting from a vertical transmission from an HSV-2-positive pregnant female to a newborn baby through intrauterine infection (5% of cases), postnatal infection (10% of cases), and perinatal infections (85% of cases) [4]. Such infection may leave the baby with long-term neurological sequelae and even cause mortality if there is no medical intervention [5]. Apart from its own pathogenicity, HSV-2 is a risk factor for human immunodeficient virus type 1 (HIV-1) acquisition, leading to a threefold increase in the chances of a patient acquiring HIV-1 [6]. A patient with both HIV-1 and HSV-2 infections has a higher likelihood of transmitting HIV-1 to another [7,8]. There have also been arguments about the relationship between HSV and human papillomavirus (HPV) and HPV-associated cervical carcinoma [9].

The high prevalence and medical expenditures of HSV-2 also emphasize the necessity for HSV-2 treatment and management. According to a systematic review [10], an estimated 491 million people aged between 14 and 49 years were suffering from HSV-2 infection in 2016 globally, constituting 13.2% of the world’s population in this age group. Women seem to be more susceptible, since 314 million people of the 491 million were female. The incidence was estimated to be 23.9 million annually. Another study in 2018 estimated that 6.35 million men and 12.2 million women in the U.S. were infected with HSV-2, and the average lifelong direct medical costs per person were USD 920 and 990, respectively, in 2018 [11,12].

Currently, antiviral therapy, which mainly involves acyclovir, valacyclovir, and famciclovir, is usually prescribed for HSV-2 infection, the goal being to alleviate symptoms or viral shedding rather than to be preventative. Once the therapy is stopped, symptoms as well as viral shedding are likely to recur. Shortcomings, such as the inability to eradicate latent viruses, a short treatment window, safety concerns in particular patients, limited prevention of viral transmission, lifelong medication, and the development of drug resistance, have not been sufficiently addressed [13,14]. Additionally, these suppressive chemicals are not effective in treating HIV-1/HSV-2 co-infectors in viral transmission [15]. Vaccines, especially therapeutic vaccines which refer to the vaccines aimed at lowering the frequency and/or severity of recurrent HSV-2 disease and/or viral DNA shedding, are regarded as the most effective and economical way to control recurrent HSV-2 disease and viral transmission with promising prospects [16]. Although much effort has been made for this purpose, there is still no licensed vaccine worldwide as of today. Fortunately, previous works on HSV-2 pathology and HSV-2 vaccine provide researchers a clue to a possible strategy for the development of a therapeutic HSV-2 vaccine, which is that a successful HSV-2 vaccine is supposed to elicit both humoral and cellular immunities harmoniously against HSV-2 [17].

In this study, based on a replication-defective Sendai virus (SeV) vector, we designed a combinatorial HSV-2 therapeutic vaccine candidate. Two different recombinant constructs comprise this vaccine candidate, that is, we inserted the genes of HSV-2-infected cell protein 27 (ICP27) and the genes of HSV-2 glycoprotein D (gD), respectively, into two replication-defective Sendai viral vectors to construct two recombinant viruses, namely, SeV-dF/HSV-2-ICP27 and SeV-dF/HSV-2-gD, to induce both cellular immunity and humoral immunities in a sufficient and balanced manner. In order to identify the immunogenicity of this vaccine candidate, we established an optimal immunization procedure and then uncovered the ability of this vaccine to activate both humoral and cellular immunities in mice. We further used an HSV-2-infected guinea pig model to evaluate the therapeutic potential of the vaccine to reduce the severity, duration, frequency, and viral shedding of recurrent HSV-2 genital disease.

## 2. Materials and Methods

### 2.1. Viruses, Cells, and Animals

Cells. Vero cell line (ATCC CCL-81) was purchased from American type culture collection (ATCC). Cells were passaged and cultured in Dulbecco’s modified essential medium (DMEM, Gibco, Grand Island, NY, USA), supplemented with 10% fetal bovine serum (FBS, Gibco, Thornton, NSW, Australia) and 1% penicillin–streptomycin (PS, Hyclone, South Logan, UT, USA). LLC-MK2/F, a modified cell line derived from the kidneys of rhesus monkeys, was a long-term storage in Weijiangbo Laboratory, National Vaccine and Serum Institute, Beijing, China. Modified essential medium (MEM, Gibco, Grand Island, NY, USA) containing 10% FBS and 1% PS was used for culturing LLC-MK2/F cells.

Recombinant virus construction and vaccine manufacture. U_L_54 gene (NCBI Gene ID:1487343) encoding HSV-2-ICP27 and U_S_6 gene (NCBI Gene ID:1487343) encoding HSV-2-gD were cloned into the plasmids containing full-length cDNA of Sendai virus. Reverse genetics was used to rescue recombinant viruses, and we successfully obtained stocks of recombinant SeVs carrying HSV-2-ICP27 and stocks of recombinant SeVs carrying HSV-2-gD, named SeV-dF/HSV-2-ICP27 and SeV-dF/HSV-2-gD, respectively. Working stocks of recombinant viruses were harvested by large-scale production in LLC-MK2/F cells and purification under GMP condition.

HSV-2 virus. HSV-2 strain VR1779 (ATCC strain designation: ATCC-2011-2) from ATCC was propagated and tittered as 2.2 × 10^7^ plaque-forming units (PFUs)/mL in Vero cells for the plaque reduction neutralizing test 50 (PRNT_50_) and viral challenge that would follow.

Animals. Female specific-pathogen-free (SPF) BALB/c mice (6–8 weeks, weighting 15–18 g) purchased from Beijing Vital River Laboratory Animal Technology Co., Ltd. (Beijing, China), were selected for the immunogenicity study. The mice were housed in SPF animal laboratory from 1 week prior to the experiment to allow them to adapt to the environment. Female SPF Hartley guinea pigs, 6–9 weeks of age, weighing 300–350 g were obtained from the same company and maintained in an animal biosafety level 2 laboratory (ABSL-2). All animal experiments and protocols had been approved by the institutional animal care and use committee of National Vaccine and Serum Institute, Beijing, China.

### 2.2. Study Design

Immunogenicity in mice. This section was separated into two parts. The first part evaluated factors that influence the vaccine’s immunogenicity, such as the dosage, the time interval between multi-doses, and the immunization strategy. All mice were vaccinated intramuscularly (i.m.) in the left thigh with allocated antigen(s)and dosage. The volume of the injection was 0.1 mL per dose. It is important to mention that in simultaneous vaccination, the volume of each component (SeV-dF/HSV-2-ICP27 and SeV-dF/HSV-2-gD) was 0.05 mL per dose, and they added up as 0.1 mL. Eight animals comprised each group. Vaccination with SeV-dF/HSV-2-ICP27 (if needed) always preceded that with its gD counterpart in light of our unpublished findings that cellular immunity triggered by SeV-dF/HSV-2-ICP27 would be inhibited by a reverse order of vaccination. Additionally, two shots of each antigen as the prime-boost procedure were required since a single shot cannot trigger a robust immune response. All animals were sacrificed 7 days after the last injection, and serum samples were drawn for the measurement of humoral immunogenicity characterized as HSV-2-specific nAb by PRNT50. At the same time, spleens were harvested for the assessment of cellular immunogenicity by IFN-γ ELISpot assay. The detailed planning of the experiments is presented in Table 1. There was a logical order to these experiments. We first determined the preferred dosage, the one with the greatest immunogenicity for each antigen, separately by comparing the immunogenicity elicited by vaccination in different dosages. It is to be noted that in dosage exploration and the following time-interval study of SeV-dF/HSV-2-gD, SeV-dF/HSV-2-gD was always administrated after prior vaccination with SeV-dF/HSV-2-ICP27. Actually, in both dosage and time-interval studies, we always determined the optimal dosage and timing for SeV-dF/HSV-2-ICP27 and then those for SeV-dF/HSV-2-gD based on the results of SeV-dF/HSV-2-ICP27. The reason is that because this vaccine candidate is designed to be combinatorial, the evaluation of SeV-dF/HSV-2-gD will not be accurate without prior vaccination with SeV-dF/HSV-2-ICP27. On the basis of optimal dosages, we then tried to identify a proper time interval between multiple doses of vaccination. Combining the appropriate dosage and time interval, we compared different immunization strategies and eventually set up an optimal immunization procedure for our combinatorial vaccine. Simultaneous vaccination included a total of four injections, and 8 logCIU for each injection was the total number of viral particles of SeV-dF/HSV-2-ICP27 and SeV-dF/HSV-2-gD mixed in equal viral units of the two components (5 × 10^7^ CIU of SeV-dF/HSV-2-ICP27 in 0.05 mL and 5 × 10^7^ CIU of SeV-dF/HSV-2-gD in 0.05 mL). The study of simultaneous vaccination was accompanied by an additional SeV-dF/HSV-2-ICP27-alone group to check whether SeV-dF/HSV-2-gD had an influence on the cellular immunogenicity of SeV-dF/HSV-2-ICP27. Similarly, the study of sequential vaccination had an SeV-dF/HSV-2-gD-immunized group as the control.

The second part was an elaborate analysis of the cellular immunogenicity of our vaccine candidates. Adopting the optimal vaccination procedure, eight BALB/c mice and their PBS-mocked controls were vaccinated and sacrificed 7 days post immunization with SeV-dF/HSV-2-ICP27. Splenocytes were isolated for the detection of IL-2/IL-4/IFN-γ secretion by ELISpot assays and for the analysis of the differentiation of the lymphocyte subpopulation and the antigen-specific T cell population by flowcytometry.

Vaccine evaluation of recurrent HSV-2 disease in a guinea pig model. Guinea pigs were used for the evaluation of the vaccine’s therapeutic effectiveness. The rationale for this substitution of mice with guinea pigs is that guinea pigs can largely replicate acute HSV-2 genital infection in humans by the vaginal HSV-2 challenge and develop spontaneous reactivation while mice cannot do so [18,19]. As a consequence, 28 female Hartley guinea pigs received the viral challenge intravaginally in the vaginal vault with 0.1 mL of a viral suspension containing 10^5^ PFUs of HSV-2 VR1779. Another 5 guinea pigs were challenged with DMEM using the same operation as the sentinels. The animals were monitored daily, and the primary episode was considered finished when genital symptoms disappeared for the first time (usually 10–14 days post the viral challenge). Any follow-up episode was recorded as a recurrence. The day on which the primary infection ended was labeled as Day 0 (D0). On the same day, the recovered guinea pigs without any HSV-2 symptoms were randomly grouped into a treatment group and a control group. The animals in the treatment group were inoculated i.m. on the left thigh on D0 and D7 with 10^8^ CIU of SeV-dF/HSV-2-ICP27 in a 100 mL dose on each day. After an interval of 14 days (on D21), 10^8^ CIU of SeV-dF/HSV-2-gD was administered and a booster on D35. For the control group, the same vaccination procedure was employed except that rather than vaccines, PBS was used for mock immunization. Daily surveillance of HSV-2 genital disease was carried out until D70, and vaginal swabs were collected every 2 days at a regular timepoint. At the end of the experiment, the guinea pigs were euthanized by intraperitoneal injection (i.p.) of 100 mg of pentobarbital sodium.

The severity of the disease was scored on a scale of 0–6, where 0 equals no sign of genital disease; 1 equals redness/swelling; 2 means the presence of 1–3 pustules; 3 means the presences of ≥4 pustules, coalescing pustules, or mild ulcers; 4 refers to the presence of severe ulcers, urine retention, or urine maceration; 5 refers to hair loss or erythema; and 6 means tissue damage with wounds. This scoring system was established from works on HSV-2 disease modeling by Stanberry LR et al. [20], the assessment for an HSV-2 therapeutic vaccine candidate [21], and the particular pathology of the HSV-2 strain VR1779.

### 2.3. Methods

ELISpot. Mouse splenocytes were isolated using a commercial mouse lymphocyte separation medium (Dakewe Biotech Co., Ltd., Shenzhen, China). Each spleen was ground gently using a sterile syringe plunger through a 70 μm cell strainer (BD Biosciences, Franklin Lake, NJ, USA) in a 35 mm Petri dish (Falcon, Corning, NY, USA) containing 5 mL of the mouse lymphocyte separation medium. The cell suspension was then transferred into a 15 mL centrifuge tube covered with 1 mL of RPMI 1640 medium (Gibco). This liquid was centrifuged at 800× *g* for 30 min and segregated into four layers from top to bottom: RPMI 1640, lymphocytes, separation medium and red blood cells, and cell debris. Lymphocytes were pipetted out slowly and washed with 10 mL of RPMI 1640. After another 10 min of centrifugation at 250× *g*, lymphocytes were resuspended, counted, and adjusted to 1 × 10^7^ cell/mL with RPMI 1640 ready for use.

IFN-γ (Cat.No.3321-2A, Mabtech, Nacka Strand, Sweden), IL-2 (Cat.No.3441-2A, Mabtech,), and IL-4 (Cat.No.3311-2A, Mabtech,) ELISpot kits were used for the detection of lymphocytes secreting IFN-γ, IL-2, and IL-4, respectively. The protocols for these three analyses were quite similar. The ELISpot plate had to be prepared 1 day before animal sacrifice. The coating antibodies (AN18 for IFN-γ, 1A12 for IL-2, and 11B11 for IL-4) were diluted to 15 μg/mL in sterile PBS at PH 7.4 and 100 μL/well of the coating antibodies was added to pre-wet PVDF plates (Millipore Corp., Burlington, MA, USA) for overnight incubation at 4 °C. On the day the animals were to be sacrificed, we activated the plates by saturating them with 200 μL/well of RPMI 1640 medium plus 10% FBS for 30 min at room temperature. The mediums were removed, and the plates were replenished with 100 μL/well of the stimuli: a nine-amino-acid peptide HGPSLYRTF in 5 μg/mL HSV-2-ICP27 (synthesized by Scilight-Peptide, Beijing, China), followed by 100 μL/well of cell suspension. A 24 h incubation allowed cytokine secretion, which was later captured by the coating antibodies. The spots were detected by the step-by-step incubation of the plates with 0.1 μg of the detection antibodies (R4-6A2-biotin for IFN-γ, 5H4-biotin for IL-2, and BVD6-24G2-biotin for IL-4), 100 μL of Streptavidin-ALP (1:1000 dilution), 100 μL of the substrate solution (BCIP/NBT) for spot development, and final spot-reading by an automated CTL immunospot analyzer and software (Immunospot, Cleveland, OH, USA).

Seroneutralization assay. The HSV-2-specific neutralizing antibody titer was determined by a PRNT_50_ test. Briefly, serum samples were twofold serially diluted in DMEM (2% FBS, 1% PS) in a 96-well plate with a final volume of 120 μL in each well and then mixed with an equal volume of HSV-2 VR1779 containing 240 PFUs of viruses. After 1 h of neutralization at 37 °C, an aliquot of 100 μL of the serum–virus mixture was transferred into 24-well plates pre-seeded with Vero cells. As positive and negative controls, respectively, 200 PFUs of viruses and DMEM were pipetted into control wells. Following 1 h of infection, the supernatant was discarded and replaced with 100 μL of MEM supplemented with 1% methyl cellulose. The plates were then maintained at 37 °C under 5% CO_2_ for 3 days. The number of plaques was counted by crystal violet indicator staining. The neutralizing antibody titer was defined as the highest dilution of the serum to reduce the formation of plaques by 50% in comparison with the plaque number in negative controls in terms of the geometry mean titer (GMT).

Flow cytometry analysis of mouse lymphocytes. Mouse splenocytes were isolated using the lymphocyte separation medium as mentioned above. For this, 100 μL/well of cell suspension adjusted to 2 × 10^7^ cells/mL was seeded into a 96-well plate. The wells were supplemented with another 100 μL/well of the solution prepared as follows: 20 μg/mL ICP27 peptide, 2 μg/mL purified anti-mouse CD28 (Cat.10210, BioLegend, San Diego, CA, USA), 2 μg/mL purified anti-mouse CD49d (Cat.103701, BioLegend), and 0.2 μL Brefeldin A solution (1000×). After 8 h of incubation of the cell-stimuli at 37 °C in 5% CO_2_ atmosphere, the cells were collected and temporarily stored in a 5 mL flow cytometry tube in a refrigerator at 4 °C. (i). For staining in the antigen-specific T cell test, the cells were washed in 2 mL of PBS and centrifuged at 350× *g* for 5 min. This was carried out twice. The supernatant was discarded, and the lymphocytes were stained using 1 mL of cell live/dead dye FVS510 (Cat. 564406, BD biosciences,) and incubated for 15 min at room temperature. Then, 2 mL of the cell staining buffer (Cat.420102, BioLegend) was added to fully stained cells. After centrifugation at 350× *g* for 5 min, the cells were resuspended in 100 μL of the staining buffer and maintained for 20 min at room temperature in dark with the fluorescent-labeled antibodies of CD4 and CD8 (PerCP anti-mouse CD4 for CD4, Cat.100538, BioLegend, and APC anti-mouse CD8 for CD8, Cat.100712, BioLegend). The cells were washed and fixed using 0.5 mL of cell fixation buffer (Cat.420801, BioLegend) for 20 min. To permeabilize cells, 2 mL of intracellular staining permeabilization buffer (Cat.401002, BioLegend) was added and the resulting liquid centrifuged twice. Next, the cells were incubated with fluorescent-labeled antibodies against cytoplasmic antigens IL-2 (FITC anti-mouse IL-2, Cat.503806, BioLegend), IL-4 (PE anti-mouse IL-4, Cat.504104, BioLegend), IFN-γ (APC-eFluor™ 780 anti-mouse IFN-y, Cat.47-7311-82, Invitrogen, Carlsbad, CA, USA), and TNF-α (PE/Cy7 anti-mouse TNF- α, Cat.47-7311-82, BioLegend) for 30 min. The cells were ready for flow cytometry analysis after being washed with 2 mL of intracellular staining permeabilization buffer and resuspended in 0.5 mL of the cell staining buffer. BD biosciences LSRII FCA instrument and FACS DIVA8.0 software were applied for tests and data analyses. (ii). For staining in the lymphocyte subpopulation test, a similar protocol was followed till the addition of fluorescent-labeled antibodies of CD4 and CD8. In lymphocyte subpopulation tests, the fluorescent-labeled antibodies used were FITC anti-mouse CD3 (Cat.100305, BioLegend), APC anti-mouse CD4 (Cat.100412, BioLegend), PE anti-mouse CD8a (Cat.100707, BioLegend), and PerCP/Cy5.5 anti-mouse CD45 (Cat.157208, BioLegend) for 20 min of incubation with cells. After being washed with 2 mL of intracellular staining permeabilization buffer and resuspended in 0.5 mL of cell staining buffer, the cells were well prepared for flow cytometry tests.

PCR quantification of HSV-2 viral shedding. For the viral shedding test, cervicovaginal swabs were collected for HSV-2 quantification by fluorescent-quantified polymerase chain reaction (qPCR). Before sample collection, a sterile cotton bud saturated with PBS was used to clean the vulva of each guinea pig. Shortly afterward, the cervicovaginal secretion of each guinea pig was sampled by slowly inserting a DMEM pre-wet swab into the vagina for about 1 centimeter and gently turning the swab for 10 rounds. The swab was then kept in a cryogenic vial containing 1 mL of DMEM (2% FBS, 1% PS) in a −80 °C environment for storage.

The viral shedding analysis was accomplished using an HSV-2 nucleic acid extraction and qPCR quantification kit (Daan Gene, Guangzhou, China). The kit has excellent specificity, sensitivity, and precision, with the range of linearity between 5 × 10^2^ copies/mL and 5 × 10^6^ copies/mL. Operations of samples strictly complied with the manufacture’s instruction provided with the kit and analyzed by an ABI Prism 7000 instrument (Thermo Fisher, Foster City, CA, USA). Known amounts of genomic HSV-2 DNA were added to delineate a standard curve for quality control and quantification, and requirements for positive and negative controls had to be satisfied to yield reliable results.

Statistics. All statistical analyses were based on the hypothesis testing performed on GraphPad Prism 9.0 software (San Diego, CA, USA). Alpha risks of 5% were set for all tests, and *p*-values lower than 0.05 were thought to suggest statistically significant differences between groups.

## 3. Results

### 3.1. Immunogenicity in Mice

In this section of our paper, we have investigated the optimal immunization procedure in a BALB/c mouse model, including the dosage, the time interval between doses, and the immunization strategy for SeV-dF/HSV-2-ICP27 and SeV-dF/HSV-2-gD. To determine an optimal dosage of SeV-dF/HSV-2-ICP27, splenocytes harvested from immunized mice were compared via IFN-γ ELISpot assay after they were administered different CIU of two doses of SeV-dF/HSV-2-ICP27. As presented in Figure 1a, immunization with SeV-dF/HSV-2-ICP27 showed that there was a dose-dependent relationship with cellular immunogenicity and that the highest immunospot count was generated by 10^8^ CIU of SeV-dF/HSV-2-ICP27, indicating the strongest cellular immunogenicity in response to a vaccination of 10^8^ CIU, as compared to that in response to 10^6^, 10^5^, and 10^4^ CIU of SeV-dF/HSV-2-ICP27 (one-way ANOVA; *p* < 0.05). There was no statistical difference between immunospot counts in response to 10^7^ and 10^8^ CIU of SeV-dF/HSV-2-ICP27 (spot count 340.54 and 407.21; *p* > 0.05). Similarly, with two doses of prior immunization with 10^8^ CIU of SeV-dF/HSV-2-ICP27, the optimal dosage for SeV-dF/HSV-2-gD was 10^8^ CIU or 10^7^ CIU, confirmed by the highest HSV-2-specific nAb analyzed by PRNT50, as shown in Figure 1b (one-way ANOVA, *p* < 0.05, as compared to 10^5^ CIU and 10^6^ CIU; no statistical difference between 10^8^ CIU and 10^7^ CIU). In view of the slightly higher titration of the neutralizing antibodies and the level of IFN-γ detection, as well as the obvious distinction between a guinea pig (for following evaluation of vaccine’s therapeutic effect) and a mouse in terms of body weight, 10^8^ CIU was justified as a proper dosage for SeV-dF/HSV-2-ICP27 and SeV-dF/HSV-2-gD for the following experiments.

The time interval between multiple doses of a vaccine is another common consideration when deciding on a vaccination procedure, including the time interval between homologous doses as prime-boost vaccinations and the time interval between heterologous combinatorial vaccinations. To determine a proper time interval between the doses of our vaccine candidate, we compared intervals of 7 days and a doubled interval between doses of SeV-dF/HSV-2-ICP27 in terms of the magnitude of cellular immunogenicity. There was no statistically significant difference between the IFN-γ immunospot counts of two different intervals (Figure 2a). A 7-day interval might be a more pragmatic choice due to the fact that vaccination at a 7-day interval led a higher spot count than that at a 14-day interval although they were not significant in statistical analysis. Moreover, the positive conversion rate in the control group was not 100% (one animal below the positive threshold, while 100% of the vaccinated mice). In addition, multi-dose vaccination with shortened intervals is more convenient and time saving for researchers to conduct experiments and for future clinical subjects to comply with.

A time-interval study of SeV-dF/HSV-2-gD was conducted after the study of SeV-dF/HSV-2-ICP27 since the evaluation of SeV-dF/HSV-2-gD will be imprecise without the preceding vaccination with SeV-dF/HSV-2-ICP27. Mice were vaccinated with two doses of SeV-dF/HSV-2-ICP27 at an interval of 7 days. The first vaccination of SeV-dF/HSV-2-gD of 10^8^ CIU was given 14 days later. The mice were vaccinated with the second dose of SeV-dF/HSV-2-gD 7 or 14 days after the first injection. Sera drawn from the mice were measured to neutralize HSV-2 viruses. The results showed that the mice that received two doses of SeV-dF/HSV-2-gD at a 14-day interval had extremely high HSV-2-neutralizing activity compared to those that received the two doses at a 7-day interval (GMT of nAb, 6.73 verses 107.63, respectively, *p*-value less than 0.01 by Mann–Whitney test), as shown in Figure 2b. Collectively, 7 days was a suitable interval between two doses of SeV-dF/HSV-2- ICP27 and 14 days for the subsequent two doses of SeV-dF/HSV-2-gD.

Simultaneous vaccination and sequential vaccination were two approaches compared in terms of vaccination strategy. Both strategies involved four doses of vaccination, one each on Day 0, Day 7, Day 21, and Day 35. The difference was that the four doses of simultaneous vaccination were exactly the same. Each dose contained overall 10^8^ CIU of viral particles with SeV-dF/HSV-2-ICP27 and SeV-dF/HSV-2-gD in a proportion of 1:1. Otherwise, the first two doses in sequential vaccination were 10^8^ CIU of SeV-dF/HSV-2-ICP27 component only and the ensuing two doses were those of SeV-dF/HSV-2-gD only. It is interesting to observe that in the group vaccinated using simultaneous strategy, there was a significantly lower spot count than that in its control group, in which mice were vaccinated with SeV-dF/HSV-2-ICP27 alone (136.17 verses 241.88; *p* < 0.05 by *t* test). It appears that simultaneous vaccination inhibited cellular immunogenicity. This phenomenon did not exist in the group vaccinated with sequential strategy, as depicted in Figure 3. Instead of being hampered, the host humoral immune response triggered by vaccination with SeV-dF/HSV-2-gD was significantly uplifted because of earlier vaccination with SeV-dF/HSV-2-ICP27 (*p* < 0.05; Mann–Whitney test).

Taken together, a combinatorial vaccination with 10^8^ CIU of SeV-dF/HSV-2-ICP27 as a primary vaccination with a homologous booster in the same dosage 7 days later, followed by two heterologous vaccinations with SeV-dF/HSV-2-gD at a time interval of 14 days between the two doses could serve as an effective immunization procedure to elicit both a robust cellular immune response and a robust humoral host immune response.

Next, we carried out a more exhaustive study of the cellular immune response induced by vaccination with SeV-dF/HSV-2-ICP27. Cytokines are effector molecules of cellular immunity, among which, IL-2, IL-4, and IFN-γ are especially crucial in human antiviral defense as they can eliminate invading viruses via direct attack or an indirect regulatory mechanism. Thus, it will be favorable if these cytokines can be adequately recruited for host anti-HSV-2 defense. To test that, ICP27-specific IL-2, IL-4, and IFN-γ secretion from mouse splenocytes after two doses of 10^8^ CIU of SeV-dF/HSV-2-ICP27 was analyzed by ELISpot assays. It was found that there was significantly greater IL-2, IL-4, and IFN-γ secretion from immunized mouse splenocytes in contrast to PBS controls (IL-2, 235.0 verses 60.8; IL-4, 420.0 verses 210.7; IFN-γ, 441.6 verses 54.7; all *p*-values less than 0.05 by *t* test), as shown in Figure 4. The data indicated that SeV-dF/HSV-2-ICP27 could boost cellular immunity against HSV-2 by generating antiviral cytokines.

Flow cytometry gave us results about T cell subset and antigen-specific T cell response (Figure 5). SeV-dF/HSV-2-ICP27-immunized mice had a promoted proportion of CD3^+^CD8^+^CTL and a downregulated proportion of CD3^+^CD4^+^Th of total T lymphocytes. The CD4/CD8 ratio was calculated as 1.546 on average, which was smaller than that of PBS-vaccinated mice (2.197; *p* < 0.05 by *t* test). The upregulation of CD8^+^ cells in vivo is generally regarded as an indicator of host antiviral reactivity. Specifically, CD8^+^ T cells are directly associated with the clearance of HSV-2 by the secretion of IFN-γ to regulate immune cells or by the direct elimination of infected cells via several pathways [22]. Albeit favorable in host antiviral defense, the unbalanced proportion of CD8^+^ T cell is not expected as it always implies a disordered status of cellular immune function. A ratio in a range between 1.5 and 2.5 is commonly considered normal for humans [23], whereas a normal value for BALB/c mice is not clear yet. It is preliminarily believed that values for humans and mice may be quite alike. One study on immune subtype distribution in BALB/c mice revealed that the CD4/CD8 ratio for male BALB/c mice was approximately 1.55 and that for female BALB/c mice was 1.38 [24]. The value is affected by several factors, such as age, habitat, infection, and intervention. Basically, the ratio in our flow cytometry analysis suggests a cemented CD4^+^ CTL response on injection of SeV-dF/HSV-2-ICP27 while a normal overall cellular immune function was not compromised.

We also employed flow cytometry to analyze antigen-specific T cell response. Comparatively, IL-2^+^CD4^+^ T cells and IFN-γ^+^CD4^+^ T cells were more greatly recalled by immunization with SeV-dF/HSV-2-ICP27 than that with PBS. With regard to antigen-specific CD8^+^ T cells, there was significantly more active recruitment of IL-2^+^CD8^+^ T cells/TNF-α^+^CD8^+^ T cells/IFN-γ^+^CD8^+^ T cells in SeV-dF/HSV-2-ICP27-vaccinated mice (all *p*-values < 0.05 by *t* test), as presented in Figure 6. The result was highly consistent with ELISpot data as described above, showing a comprehensive cellular immunogenicity activated by SeV-dF/HSV-2-ICP27.

### 3.2. Vaccine Evaluation in a Guinea Pig Model of Recurrent HSV-2 Disease

Twenty-eight female guinea pigs were intravaginally inoculated with HSV-2 VR1779 at 10^5^ PFUs per animal. Primary infection, with typical HSV-2-associated symptoms and lesions, that resembles the course of human infection arose around the genital tract of twenty-seven animals. The guinea pig that failed to establish primary infection (or the primary infection may have occurred in form of subclinical infection) was euthanized by 100 mg of pentobarbital sodium i.p. Fourteen days after viral challenge, symptoms of genital disease in all animals disappeared, indicating that the primary episode had ended. To properly designate guinea pigs into treatment or PBS control groups, the severity of disease in every individual was calculated based on the 0–6 scoring system. Guinea pigs with the same score were assembled together and then distributed equally into treatment and control groups. In this way, an equivalent background of disease condition was ensured between the two groups to eradicate the baseline difference in their recurrent disease. Eventually, 14 guinea pigs were put in the treatment group, while 13 were put in the PBS control group. On the same day, noted as Day 0 (D0), immunization of the guinea pigs began with one dose of 10^8^ CIU of SeV-dF/HSV-2-ICP27 for the treatment group animals and PBS for the control animals. Another vaccination of 10^8^ CIU of SeV-dF/HSV-2-ICP27 was administrated on Day 7 (D7), followed by two injections of 10^8^ CIU of SeV-dF/HSV-2-gD, one administered on Day 21 (D21) and one on Day 35 (D35). The mice were kept under daily observation and swabbing was performed every other day till Day 70 (D70). Disease severity, disease duration, the frequency of recurrent disease, and viral shedding were analyzed based on observations and vaginal secretion samples. Note that the sentinel animals challenged with DMEM remained in good condition without manifest appearance of any kind of infection during the whole observation. This proved that any symptoms that occurred in the observed animals were incurred restrictively by HSV-2 infection.

Measuring severity involves measuring the degree of the recurrent genital lesions in guinea pigs. To evaluate the severity, the everyday scores for each guinea pig throughout the 70-day observation were summed up and the total scores of each individual in the two groups were compared. In terms of each individual, guinea pigs in the treatment group had an average summed score of 7.57, which was significantly lower (58.66%) than the average score of 18.31 for guinea pigs in the control group (*p*-value < 0.05; Mann–Whitney test) (Figure 7a). A more detailed graphic illustration widely used in publications associated with the evaluation of HSV-2 vaccine involves depicting a curve of cumulative scores that delineates a cumulative trend of disease severity. The cumulative score is created by plotting the everyday average score plus the summed average score of preceding day(s) on the *y*-axis against days post first vaccination, from D0 to D70, on the *x*-axis in a coordinate system, as illustrated in Figure 7b. A Wilcoxon matched-pairs test inferred that the cumulative score of the treatment group was significantly lower than that of the PBS group.

The duration of recurrent disease refers to the sum of days on which a guinea pig displays symptoms of HSV-2 genital infection. In the entire course of observation spanning 70 days, vaccinated animals went through an overall 3.87-day-long recurrence per animal, in contrast to 7.08 days for the control animals, as shown in Figure 7c. A 0.045 *p*-value derived from a Mann–Whitney U test indicated a significant difference between the two groups. In other words, immunization with SeV-dF/HSV-2-ICP27 and SeV-dF/HSV-2-gD managed to shorten the duration of recurrent HSV-2 disease in guinea pigs. A significant difference also existed in the frequency of recurrence. We define one recurrence as an episode of recurrent disease that occurs with distinguishable HSV-2 genital manifestation, from its appearance to its full clearance. A new recurrence is accepted when there is no presence of any symptoms the day before. In a total of 70-day period from the first vaccination, animals in the treatment group were observed to experience an average of 2.07 episodes of recurrence, while for animals in the PBS control group, this value was 3.77, as demonstrated in Figure 7d. A 45.09% reduction in the frequency of recurrence was achieved (*p* = 0.021; Mann–Whitney U test). It is noteworthy that all guinea pigs in the control group suffered from recurrent disease, while only 64.29% (five) of the guinea pigs in the treatment group had recurrent episodes. Some individuals were perfectly protected from recurrent HSV-2 disease as a result of treatment with SeV-dF/HSV-2-ICP27 and SeV-dF/HSV-2-gD.

The vaccine also showed an inhibitory effect on cervicovaginal HSV-2 viral shedding. During the entire experiment, 605 samples of cervicovaginal swabs were taken from animals in the treatment group. Of these swabs, 1.98% (12) were tested to be HSV-2 positive by HSV-2 real-time fluorescent quantitative PCR. In comparison, 4.54% (23) of the swabs of the 570 samples collected from the animals in the control group were found to be HSV-2 positive. A single positive swab suggests that HSV-2 viral shedding occurred in the guinea pig being swabbed on the day of sampling and intervention (vaccination) failed to prevent the shedding. The difference in viral shedding between treatment and control groups was significant, with a *p*-value equaling 0.039 by chi-square test, which indicates that guinea pigs vaccinated with SeV-dF/HSV-2-ICP27/SeV-dF/HSV-2-gD had a lower frequency of viral shedding than the PBS control group.

## 4. Discussion

In this study, we present an original vaccine formulation, replication-defective recombinant Sendai viruses conveying HSV-2 envelope protein gD and non-structural protein ICP27 for the treatment of recurrent HSV-2 disease. The underlying concept of this vaccine is to trigger both cellular and humoral immune response sufficiently, which is proven to be feasible by our results. A new antigenic combination, a novel delivery system for the HSV-2 vaccine with the ability to trigger antiviral CD4^+^ and CD8^+^ T cell-mediated immunity in addition to humoral immunity is proposed as there are major gaps in the present HSV-2 vaccine design proposals [25] and here we try to bridge these gaps.

In a BALB/c mouse model, two injections of 10^8^ CIU of SeV-dF/HSV-2-ICP27 each and another two SeV-dF/HSV-2-gD injections in the same dose could elicit ICP27-specific IFN-γ secretion and HSV-2-specific neutralizing antibody generation. Our in-depth research into the cellular immunogenicity of SeV-dF/HSV-2-ICP27 revealed that IL-2-, TNF-α-, and IFN-γ-antigen-specific CTLs and IL-2- and IFN-γ-antigen-specific Th cells were successfully recalled by two infections of SeV-dF/HSV-2-ICP27 and these cells jointly contributed to an in vivo increase in IL-2, TNF-α, and IFN-γ levels. These cytokines play significant roles in human antiviral defense. The major physiological function of IL-2 and IL-4 is to regulate the immune system, especially to interact with B cells [16,26,27]. Such regulation was evidenced by our results in that the neutralizing antibody titer generated by SeV-dF/HSV-2-gD was promoted by prior immunization with SeV-dF/HSV-2-ICP27. IFN-γ, a member of the type 2 interferon family, is also a crucial player in human antiviral protection, orchestrating numerous immune pathways and responses. IFN-γ facilitates antigen processing and presentation by mediating MHC II molecules, such as macrophages, and upregulates the activities of NK cells, B cells, and other immunocytes. IFN-γ, together with type 1 IFN, can directly attack inbreaking viruses by inducing the generation of various antiviral proteins, leading to abnormal transcription of viral RNA and their eventual death in the very early stages of viral infection [28,29]. Additionally, IFN-γ had an inhibitory effect on the reactivation of latent HSV in sacral trigeminus [30,31]; CD4+ T-specific IFN-γ could shield the mice from the lethal dose challenge of HSV-2 [32]. In humans, CD8+T-specific IFN-γ and several co-effectors could jointly maintain the quiescent state of HSV infection [33]. In the course of HSV infections, several viral proteins can launch a shutdown of IFN activities, whereby HSV-2 can successfully start the invasion and establishment of latency [34]. But what if IFN can get normally engage in the antiviral defense? It will be a possible strategy that is worth trying.

In a recurrent HSV-2 disease model in our study, the same vaccination strategy was applied to HSV-2-infected guinea pigs. It is noteworthy that our vaccine not only lowered the severity, duration, and frequency of the recurrent genital disease by 58.66%, 45.34%, and 45.09%, respectively, compared to the PBS control group but also reduced viral shedding by 50.99%. The reduction in viral shedding and the consequently reduced viral transmission from infected individuals to seronegative persons are always recognized as a great social benefit for public health. All these outcomes show that our selection of antigens and delivery system is appropriate and immunization with SeV-dF/HSV-2-gD and SeV-dF/HSV-2-ICP27 can be a promising treatment for recurrent HSV-2 disease.

The most intensively used antigen of choice in HSV-2 vaccine design and also one of the antigens in our vaccine design is gD. It is one of the 13 glycoproteins on the HSV-2 viral envelope and an essential glycoprotein required for the binding of viral particles to the receptors on target cells. The binding alters the configuration of the cell surface protein, which enables gB, gH, gL, and several glycoproteins to perform their synergetic functions for viral fusion and viral entry [35]. As gD is the most abundant protein on the surface of HSV-2-infected cell, it is likely to be recognized by the host immune system and initiate antiviral defense. In fact, the administration of a gD-based vaccine could stimulate HSV-2-specific neutralizing antibody generation in animal models and humans. Neutralizing antibodies are a requisite participant in the repression of HSV-2 infection since they are responsible for blocking the site of viral binding and retarding viral diffusion [36]. Supportive outcomes of vaccine candidates, such as subunit gD2ΔTMR/ICP4.2 GEN-003 (by Genocea, Cambridge, MA, USA) [21,37,38], gD DNA vaccine COR-1 (by Admedus Ltd., now Anteris Technologies Ltd., Toowong, QLD, Australia) [39,40], gD/gB subunit vaccine (by Chiron Corp., Emeryville, CA, USA) [41] and adjuvanted gD vaccine (by GlaxoSmithKline, Brentford, UK) [42] in pre-clinical tests and clinical trials have confirmed the efficacy of gD in eliciting neutralizing antibodies, but cellular immunity has not been intensively activated in humans albeit it is successful in animal tests. A widely accepted explanation is that in humans, gD is capable of inducing humoral immunity but not cellular immunity, both of which are collectively required to be well functioning for human anti-HSV-2 immunity.

To address this issue, we chose ICP27 as an additional antigen in our vaccine design. ICP27 is a phosphoprotein encoded by HSV-2 U_L_54 and one of the very first proteins expressed in the course of viral infection in host cells. As an immediate early protein, ICP27 acts as an essential regulator in viral activities, especially during early-phase activities. One of the most important functions that ICP27 performs is that it mediates the repression of host cell mRNA splicing and consequently the shutdown of protein synthesis, including the inhibition on interferon generation [43,44]. This process contributes to viral evasion from host immune surveillance. In theory, a previous exposure to ICP27 might uplift quick recognition and strong immune response to ICP27 to limit viral invasion at an early phase. Moreover, ICP27 is capable of facilitating host cellular immunity, particularly cytotoxic lymphocyte (CTL) response and IFN-γ secretion. CTL destroys HSV-2-infected cells prior to viral late gene expression, so the release of infectious progenies is prevented and neighboring cells are well protected from cell-to-cell transfection [45]. The rationale for selecting ICP27 is clear, and efforts have been made for such purpose. Bright H. et al. constructed prophylactic HSV-2 DNA vaccines encoding gB and gD, and the protection was largely promoted by the addition of ICP27-expressing DNA vaccine [46]. In this study, gB and gD were shown to be immunogenic for humoral immunity, while ICP27 elicited an antiviral T cell response. A study by Rebecca J. et al. also demonstrated that immunization with ICP27 (DNA vaccine) could reduce viral shedding and alleviate clinical manifestation of HSV-2 infection in a mouse model [47].

The delivery system for exogenous genes is another determinant for vaccine performance, for which our choice was the Sendai virus. SeV, or murine parainfluenza virus type 1, is an infectious agent for murine pneumonia belonging to the Respirovirus genus in the Paramyxoviridae family, with a non-segment, single-stranded, negative-sense RNA genome. Safety, risk of viral shedding, and pre-existing immunity are major concerns for a potential viral vector for gene therapy and vaccine development. For decades, SeV’s safety profile, high expression of foreign genes, and broad host spectrum have made it an attractive option. No case of human SeV infection has been reported since its first discovery, and SeV is accepted as a murine pathogen but not a human pathogen. SeV completes its entire life cycle in the cytoplasm without the nucleus phase, thereby eliminating the risk of gene integration into the host chromosomes [48]. What makes the SeV vector more reliable is that we endowed SeV with a replication-deficient feature, achieved by knocking out F from the SeV Z strain backbone. LLC/MK2 cells provide the F gene in trans for F-gene-deleted SeV-dF to package into an infectious particle in the cell culture system so that the vectors can be amplified for large-scale production. Such a mechanism will not work in cell lines that do not supply the F gene. Thus, the progeny viruses will become transmission defective. The safety of SeV-dF has been confirmed by our preclinical toxicological and pharmacokinetic assessment. No adverse event was observed in BALB/c mice and rhesus macaques after a total of four inoculations with SeV-dF. Apart from detectable copies of SeV in blood on Day 3 after immunization, SeV was not detected in the main organs (lungs, the airway, kidneys, the liver, and intestines) in rhesus macaques and in animal excretions (feces, urine, and nasal secretion), indicating the great safety profile and the fact that there is no hazard of environmental contamination. The results from other clinical trials in which SeV served as a vaccine against HIV-1, human parainfluenza virus type 1 (hPIV-1), and several pathogens also verify its satisfying safety [49,50].

Another issue of concern is the pre-existing human immunity against SeV. Although SeV is not an infectious agent for humans, it belongs to the Paramyxoviridae family, along with hPIV-1, which causes human respiratory disease and populations worldwide are susceptible to it. SeV is closely related to hPIV-1 in genetic sequence and viral structure due to their homology. They have 72% similarity in their nucleic acid sequence, specifically 83% in N, 53% in P, 87% in M, 68% in F, 72% in HN, and 86% in L, which are six transcription units shared by both SeV and hPIV-1, and 53–87% similarity in the amino acid sequence [51,52,53,54,55,56]. So, the concerns about cross-reactivity are reasonable. The antibodies induced by any previous hPIV-1 infection may cross-react with the SeV vector. Fortunately, in spite of the fact that cross-reactivity does exist, the degree of such cross-reactivity characterized by antibody levels against SeV is relatively low, suggesting that pre-existing immunity may not pose a serious problem. One research enrolling 255 individuals from Africa, Europe, the U.S., and Japan illustrated that 92.5% of these individuals had neutralizing antibodies against SeV in their serum samples, with a median titer at 60.6. The majority of the subjects (71.7%) had titers lower than 100, while only a few had titers greater than 1000 [57]. This magnitude of pre-existing antibodies may not affect the efficacy of the SeV vector. To date, the attempts to use SeV as a tool for gene or antigen transfer in clinical trials have yielded favorable results in the absence of SeV antibodies in human subjects, implying the trivial impact of pre-existing immunity against the SeV vector. Several strategies, including the primary-boost procedure [58] and a change of the immunization route [59], have been tested to surmount the pre-existing immunity. To this end, we determined two injections of SeV-dF/HSV-2-gD or SeV-dF/HSV-2-ICP27 as the booster shot and sequential vaccination with SeV-dF/HSV-2-ICP27 following SeV-dF/HSV-2-gD as our combinatorial vaccination schedule.

## 5. Conclusions

In a nutshell, there is considerable evidence that our Sendai virus vectors carrying HSV-2-gD and HSV-2-ICP27 genes could induce high levels of both cellular and humoral immunity in mice and significantly reduce the severity, duration, and frequency of recurrent HSV-2 genital disease in guinea pigs, with a decrease in viral shedding. This combinatorial vaccine candidate is preliminarily believed to meet the requirements of an effective therapeutic HSV-2 vaccine. More studies, for example, on long-term efficacy and safety, are in progress with great potential.

## Figures and Tables

**Figure 1 vaccines-11-01752-f001:**
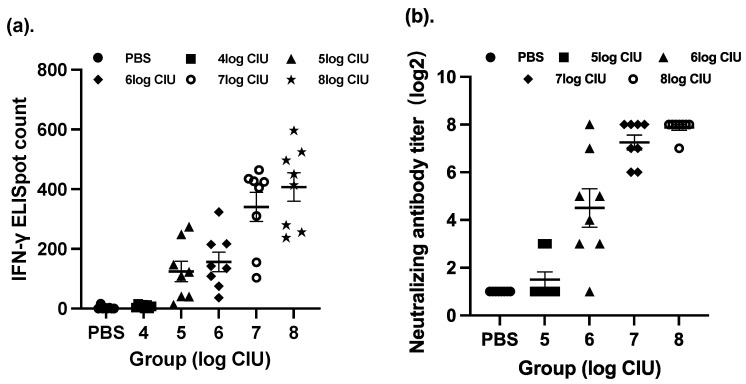
Immune responses induced by vaccination with SeV-dF/HSV-2-ICP27 and SeV-dF/HSV-2-gD. Each single point represents the measurement of each individual, and the error bar reflects the mean with a standard error of measurement (SEM) within each group. (**a**) The mice were immunized intramuscularly with two doses of SeV-dF/HSV-2-ICP27 of 10^4^, 10^5^, 10^6^, 10^7^, and 10^8^ CIU or PBS as the control. The figure represents the IFN-γ ELISpot immunospot counts of mouse splenocytes from BALB/c mice vaccinated with different dosages of SeV-dF/HSV-2-ICP27. (**b**) The mice were administered two doses of SeV-dF/HSV-2-ICP27 and two subsequent doses of SeV-dF/HSV-2-gD of 10^5^, 10^6^, 10^7^, and 10^8^ CIU or PBS as the control. Serum samples were collected, and HSV-2-specific neutralizing antibodies were analyzed by PRNT50. Titers in Figure (**b**) are illustrated in log2 form.

**Figure 2 vaccines-11-01752-f002:**
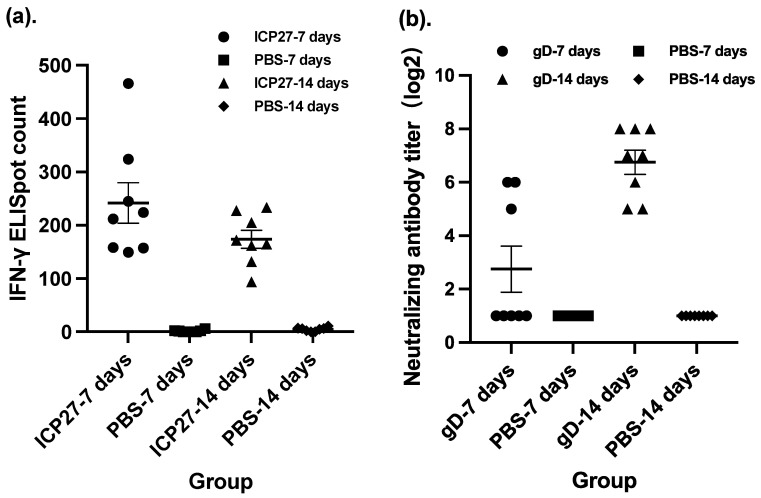
Different time intervals between injections led to different levels of immunogenicity. ICP27: vaccination with SeV-dF/HSV-2-ICP27; gD: vaccination with SeV-dF/HSV-2-gD; PBS: vaccination with PBS. (**a**) Cellular immunogenicity featured by the IFN-γ immunospot count of mouse splenocytes after vaccination with two doses of SeV-dF/HSV-2-ICP27 at different time intervals in BALB/c mice. (**b**) Humoral immunogenicity characterized by the serum HSV-2-neutralizing antibody titer after vaccination with SeV-dF/HSV-2-gD at a 7-day or a 14-day interval.

**Figure 3 vaccines-11-01752-f003:**
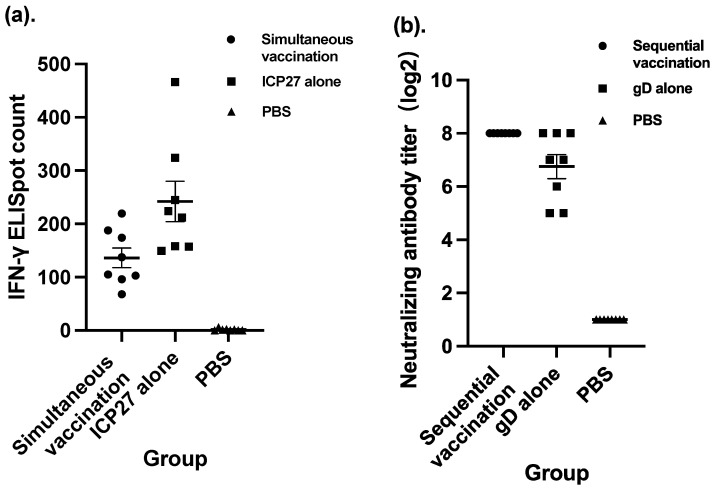
Comparison between simultaneous vaccination and vaccination with SeV-dF/HSV-2-ICP27 alone, and comparison between sequential vaccination and vaccination with gD alone. ICP27: vaccination with SeV-dF/HSV-2-ICP27; gD: vaccination with SeV-dF/HSV-2-gD; PBS: vaccination with PBS. (**a**) Cellular immunogenicity of simultaneous vaccination compared to that of SeV-dF/HSV-2-ICP27 vaccination alone. Simultaneous vaccination involved four doses of vaccination consisting of equal amounts of SeV-dF/HSV-2-ICP27 and SeV-dF/HSV-2-gD, each dose comprising 10^8^ CIU of total viral particles, while the mice in the ICP27 control group were vaccinated with four doses of 10^8^ CIU of SeV-dF/HSV-2-ICP27 alone. (**b**) Humoral immunogenicity of sequential vaccination and SeV-dF/HSV-2-gD vaccination. Sequential vaccination is defined as two doses of 10^8^ CIU of SeV-dF/HSV-2-ICP27 and two subsequent doses of 10^8^ CIU of SeV-dF/HSV-2-gD. The mice in the control group were immunized by SeV-dF/HSV-2-gD alone.

**Figure 4 vaccines-11-01752-f004:**
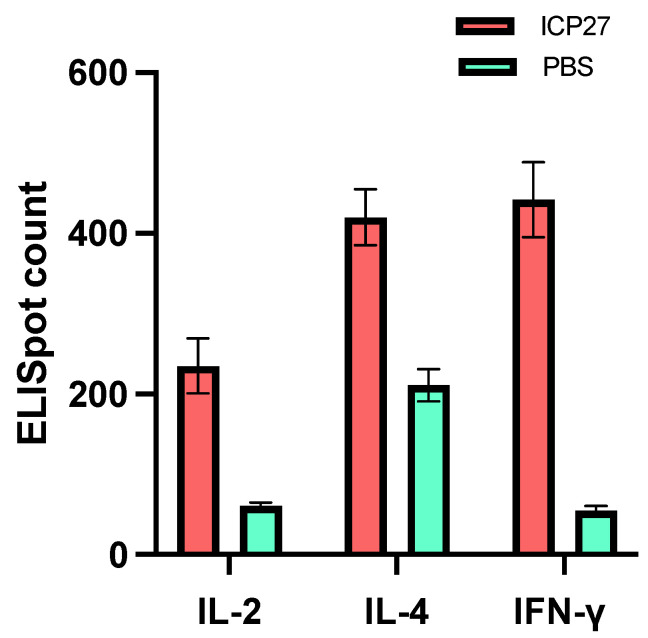
Secretion of IL-2, IL-4, and IFN-γ from mouse splenocytes after immunization with SeV-dF/HSV-2-ICP27 examined by ELISpot assay. ICP27: vaccination with SeV-dF/HSV-2-ICP27; PBS: vaccination with PBS. After two doses of 10^8^ CIU of SeV-dF/HSV-2-ICP27, the immunized mice produced a sufficient cellular immune response to viral stimuli, characterized by IL-2, IL-4, and IFN-γ generation.

**Figure 5 vaccines-11-01752-f005:**
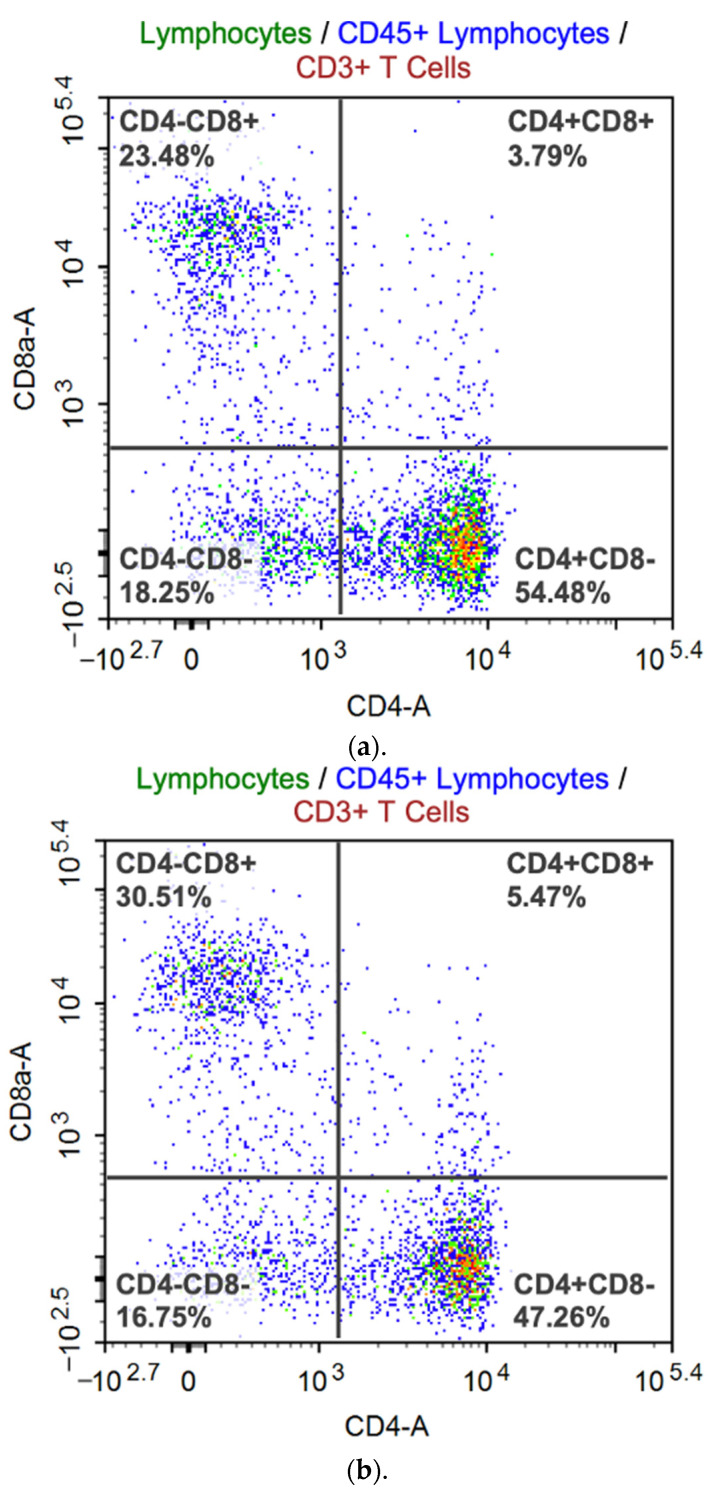
Representative flow cytometry analysis. Dots represent the distribution of the CD3^+^ subgroup in four quadrants. Red dot: singlets; green dot: lymphocytes; blue dots: CD45+ lymphocytes; wine red dots: CD3+ T cells; yellow dots: CD4 + T cells. Cells in the upper-left quadrant, CD4 negative and CD8 positive; cells in the upper-right quadrant, CD4 and CD8 positive; cells in the lower-left quadrant, CD4 and CD8 negative; and cells in the lower-right quadrant, CD4 positive and CD8 negative. The percentages provided in the figures are the frequencies of the distributed cell subtypes of CD3^+^ lymphocytes, and CD4/CD8 ratios are results of the division of the percentage in the lower-right quadrant by that in the upper-left quadrant. (**a**) A representative flow cytometry diagram of a BALB/c mouse vaccinated with two doses of 10^8^ CIU of SeV-dF/HSV-2-ICP27. (**b**) Representative flow cytometry data of a PBS-immunized BALB/c mouse.

**Figure 6 vaccines-11-01752-f006:**
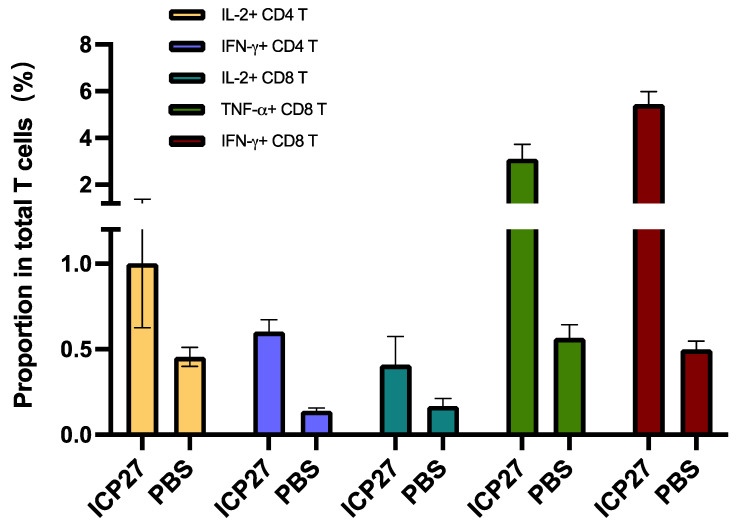
Antigen-specific T cell examination by flow cytometry. ICP27: vaccination with SeV-dF/HSV-2-ICP27; PBS: vaccination with PBS. Mice received two doses of 10^8^ CIU of SeV-dF/HSV-2-ICP27 or PBS and were sacrificed for spleen harvest and further flow cytometry. The proportions shown in the figure are the frequencies of each type of antigen-specific T cells in total T cells. SeV-dF/HSV-2-ICP27-vaccinated mice had higher splenic IL-2^+^CD4^+^ T cells, IFN-γ^+^CD4^+^ T cells, IL-2^+^CD8^+^ T cells, TNF-α^+^CD8^+^ T cells, and IFN-γ^+^CD8^+^ T cells.

**Figure 7 vaccines-11-01752-f007:**
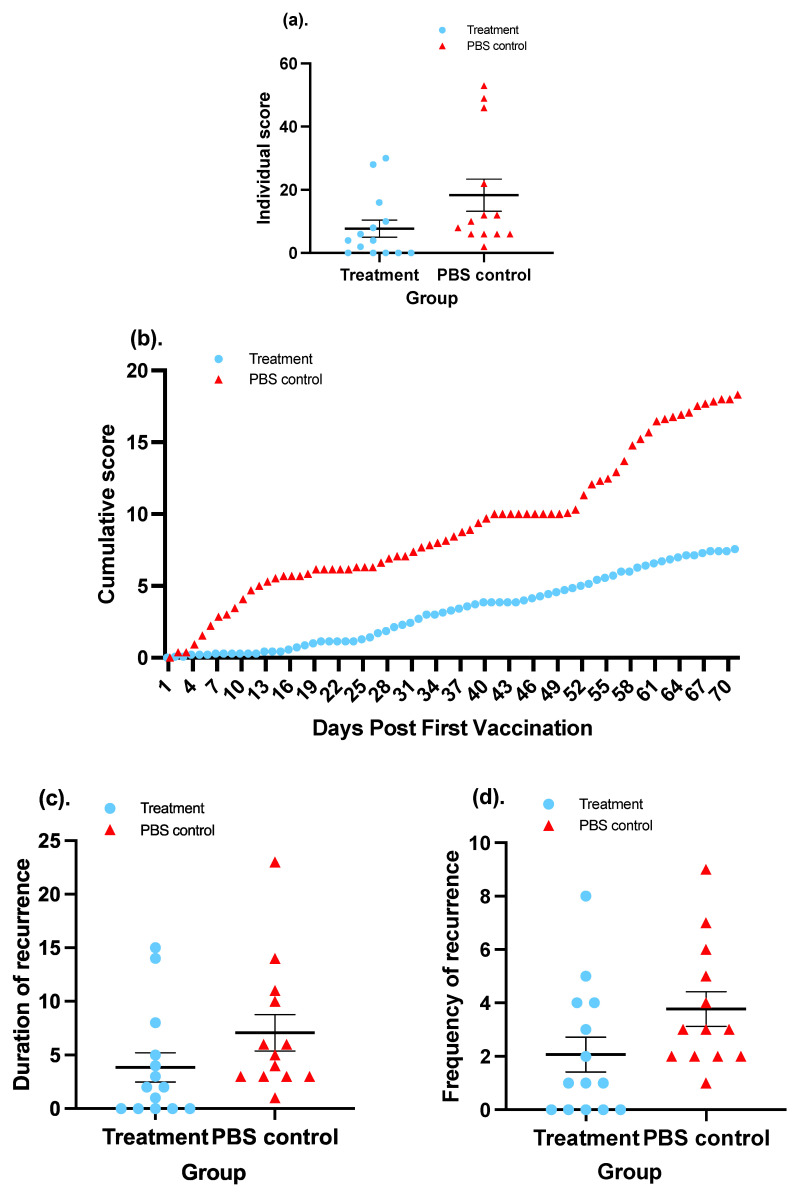
Evaluation of the vaccine candidate in terms of therapeutic effect on recurrent HSV-2 disease in a guinea pig model. Treatment: The animals in the treatment group received two doses each of 10^8^ CIU of SeV-dF/HSV-2-ICP27 i.m. at a 7-day interval, followed by two doses of 10^8^ CIU of SeV-dF/HSV-2-gD 14 days apart. Control animals were injected with PBS on the same days. (**a**) Cumulative scores for each individual guinea pig in both groups. (**b**) Daily average cumulative scores in a 70-day period. (**c**) Duration of recurrence in guinea pigs in different groups. (**d**) Frequency of recurrence in guinea pigs in two groups.

**Table 1 vaccines-11-01752-t001:** Arrangement of experiments for the development of a proper immunization procedure.

Factors	Grouping (−logCIU)	Vaccination at				
		Day0	D7	D14	D21	D35
Dosage (SeV-dF/HSV-2-ICP27)	4, 5, 6, 7, 8, PBS	√	√	/	/	/
Dosage (SeV-dF/HSV-2-gD)	5, 6, 7, 8, PBS	√	√	/	√	√
Time interval: 7 days (SeV-dF/HSV-2-ICP27)	8, PBS	√	√	/	/	/
Time interval: 14 days (SeV-dF/HSV-2-ICP27)	8, PBS	√	/	√	/	/
Time interval: 7 days (SeV-dF/HSV-2-gD)	8, PBS	√	√	√	√	/
Time interval: 14 days (SeV-dF/HSV-2-gD)	8, PBS	√	√	/	√	√
Strategy: Simultaneous vaccination	8, PBS	√	√	/	√	√
Strategy: Sequential vaccination	8, PBS	√	√	/	√	√

## Data Availability

All data used in this article are available from the corresponding author by request.

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
