# Peer review of "Immunogenicity and Therapeutic Efficacy of a Sendai-Virus-Vectored HSV-2 Vaccine in Mouse and Guinea Pig Models"

_vaccines, 2023, doi:10.3390/vaccines11121752_

Round 1

Reviewer 1 Report

Comments and Suggestions for Authors

In this manuscript, Ren et al. describe a combinatorial vaccination strategy designed to induce robust humoral and cellular immune responses. The vaccine is based on Sendai virus (SeV), but how the vaccine was constructed is not mentioned. The authors optimized the vaccination dose, timing and combination of the two vectors and showed that vaccination did indeed induce some neutralizing antibodies against gD and ICP27-specific CD8 responses. Furthermore, they show that this strategy could reduce the frequency of HSV-2 recurrence and disease severity in latently infected guinea pigs. Although the beneficial effects in animal models are encouraging, the authors should compare the immune responses between the vaccinated and control groups to show that the therapeutic effects are due to vaccine-induced anti-HSV-2 immune responses.

1. The rationale for selecting ICP27 and gD for the anti-HSV-2 vaccine is not clear.

2. The construction of the SeV vector is not described.

3. The legend of Figure 1 is poorly written and misleading.

4. Statistical analysis is needed for Figures 6 and 7.

5. In the guinea pig model, the authors should compare the neutralizing antibody titers and HSV-2-specific CD8 responses between the vaccinated and control groups.

    Comments on the Quality of English Language

The manuscript is not well written and requires extensive language editing.

Reviewer 2 Report

Comments and Suggestions for Authors

The field of HSV-2 virus vaccination should be properly described as the prior finding might change your explanations and data analysis. Your paper has missed out just about most significant publications as listed below. Please see if you can rewrite your introduction with proper recognition to prior work and then reanalyze your data based on any previous data along with an explanation of the outcome analysis.

 De Rose DU, Bompard S, Maddaloni C, Bersani I, Martini L, Santisi A, Longo D,

Ronchetti MP, Dotta A, Auriti C. Neonatal herpes simplex virus infection: From

the maternal infection to the child outcome. J Med Virol. 2023 Aug;95(8):e29024.

doi: 10.1002/jmv.29024. PMID: 37592873.

 Sausen DG, Shechter O, Gallo ES, Dahari H, Borenstein R. Herpes Simplex

Virus, Human Papillomavirus, and Cervical Cancer: Overview, Relationship, and

Treatment Implications. Cancers (Basel). 2023 Jul 20;15(14):3692. doi:

10.3390/cancers15143692. PMID: 37509353; PMCID: PMC10378257.

 Preda M, Manolescu LSC, Chivu RD. Advances in Alpha Herpes Viruses Vaccines

for Human. Vaccines (Basel). 2023 Jun 12;11(6):1094. doi:

10.3390/vaccines11061094. PMID: 37376483; PMCID: PMC10302067.

 Sharma D, Sharma S, Akojwar N, Dondulkar A, Yenorkar N, Pandita D, Prasad SK,

Dhobi M. An Insight into Current Treatment Strategies, Their Limitations, and

Ongoing Developments in Vaccine Technologies against Herpes Simplex Infections.

Vaccines (Basel). 2023 Jan 17;11(2):206. doi: 10.3390/vaccines11020206. PMID:

36851084; PMCID: PMC9966607.

 Chentoufi AA, Dhanushkodi NR, Srivastava R, Prakash S, Coulon PA, Zayou L,

Vahed H, Chentoufi HA, Hormi-Carver KK, BenMohamed L. Combinatorial Herpes

Simplex Vaccine Strategies: From Bedside to Bench and Back. Front Immunol. 2022

Apr 25;1849515. doi: 10.3389/fimmu.2022.849515. PMID: 35547736; PMCID:

PMC9082490.

 Krishnan R, Stuart PM. Developments in Vaccination for Herpes Simplex Virus.

Front Microbiol. 2021 Dec 7;1798927. doi: 10.3389/fmicb.2021.798927. PMID:

34950127; PMCID: PMC8691362.

 Wijesinghe VN, Farouk IA, Zabidi NZ, Puniyamurti A, Choo WS, Lal SK. Current

vaccine approaches and emerging strategies against herpes simplex virus (HSV).

Expert Rev Vaccines. 2021 Sep;20(9):1077-1096. doi:

10.1080/14760584.2021.1960162. Epub 2021 Aug 9. PMID: 34296960.

 Stanfield BA, Kousoulas KG, Fernandez A, Gershburg E. Rational Design of

Live-Attenuated Vaccines against Herpes Simplex Viruses. Viruses. 2021 Aug

18;13(8):1637. doi: 10.3390/v13081637. PMID: 34452501; PMCID: PMC8402837.

 AlMukdad S, Harfouche M, Wettstein A, Abu-Raddad LJ. Epidemiology of herpes

simplex virus type 2 in Asia: A systematic review, meta-analysis, and meta-

regression. Lancet Reg Health West Pac. 2021 Jun 9;1100176. doi:

10.1016/j.lanwpc.2021.100176. PMID: 34527970; PMCID: PMC8356094.

Aschner CB, Herold BC. Alphaherpesvirus Vaccines. Curr Issues Mol Biol.

2021;4469-508. doi: 10.21775/cimb.041.469. Epub 2020 Sep 23. PMID: 32963118;

PMCID: PMC8211365.

Egan K, Hook LM, LaTourette P, Desmond A, Awasthi S, Friedman HM. Vaccines

to prevent genital herpes. Transl Res. 2020 Jun;220:138-152. doi:

10.1016/j.trsl.2020.03.004. Epub 2020 Mar 16. PMID: 32272093; PMCID: PMC7293938.

Sandgren KJ, Truong NR, Smith JB, Bertram K, Cunningham AL. Vaccines for

Herpes Simplex: Recent Progress Driven by Viral and Adjuvant Immunology. Methods

Mol Biol. 2020;2060:31-56. doi: 10.1007/978-1-4939-9814-2_2. PMID: 31617171.

Schiffer JT, Gottlieb SL. Biologic interactions between HSV-2 and HIV-1 and

possible implications for HSV vaccine development. Vaccine. 2019 Nov

28;37(50):7363-7371. doi: 10.1016/j.vaccine.2017.09.044. Epub 2017 Sep 25. PMID:

28958807; PMCID: PMC5867191.

Gottlieb SL, Giersing B, Boily MC, Chesson H, Looker KJ, Schiffer J,

Spicknall I, Hutubessy R, Broutet N; WHO HSV Vaccine Impact Modelling Meeting

Working Group. Modelling efforts needed to advance herpes simplex virus (HSV)

vaccine development: Key findings from the World Health Organization

Consultation on HSV Vaccine Impact Modelling. Vaccine. 2019 Nov

28;37(50):7336-7345. doi: 10.1016/j.vaccine.2017.03.074. Epub 2017 Jun 21. PMID:

28647165.

Gottlieb SL, Johnston C. Future prospects for new vaccines against sexually

transmitted infections. Curr Opin Infect Dis. 2017 Feb;30(1):77-86. doi:

10.1097/QCO.0000000000000343. PMID: 27922851; PMCID: PMC5325242.

Johnston C, Gottlieb SL, Wald A. Status of vaccine research and development

of vaccines for herpes simplex virus. Vaccine. 2016 Jun 3;34(26):2948-2952. doi:

10.1016/j.vaccine.2015.12.076. Epub 2016 Mar 11. PMID: 26973067.

Kaufmann JK, Flechtner JB. Evolution of rational vaccine designs for genital

herpes immunotherapy. Curr Opin Virol. 2016 Apr;180-86. doi:

10.1016/j.coviro.2016.01.021. Epub 2016 Feb 18. PMID: 26896782.

Awasthi S, Shaw C, Friedman H. Improving immunogenicity and efficacy of

vaccines for genital herpes containing herpes simplex virus glycoprotein D.

Expert Rev Vaccines. 2014 Dec;13(12):1475-88. doi: 10.1586/14760584.2014.951336.

Epub 2014 Aug 20. PMID: 25138572.

Kuo T, Wang C, Badakhshan T, Chilukuri S, BenMohamed L. The challenges and

opportunities for the development of a T-cell epitope-based herpes simplex

vaccine. Vaccine. 2014 Nov 28;32(50):6733-45. doi:

10.1016/j.vaccine.2014.10.002. Epub 2014 Oct 16. PMID: 25446827; PMCID:

PMC4254646.

Awasthi S, Friedman HM. Status of prophylactic and therapeutic genital

herpes vaccines. Curr Opin Virol. 2014 Jun;6-12. doi:

10.1016/j.coviro.2014.02.006. Epub 2014 Mar 12. PMID: 24631871.

Coleman JL, Shukla D. Recent advances in vaccine development for herpes

simplex virus types I and II. Hum Vaccin Immunother. 2013 Apr;9(4):729-35. doi:

10.4161/hv.23289. Epub 2013 Feb 26. PMID: 23442925; PMCID: PMC3903888.

Dropulic LK, Cohen JI. The challenge of developing a herpes simplex virus 2

vaccine. Expert Rev Vaccines. 2012 Dec;11(12):1429-40. doi: 10.1586/erv.12.129.

PMID: 23252387; PMCID: PMC3593236.

Johnston C, Koelle DM, Wald A. HSV- in pursuit of a vaccine. J Clin

Invest. 2011 Dec;121(12):4600-9. doi: 10.1172/JCI57148. Epub 2011 Dec 1. PMID:

22133885; PMCID: PMC3223069.

Kemble G, Spaete R. Herpes simplex vaccines. In: Arvin A, Campadelli-Fiume

G, Mocarski E, Moore PS, Roizman B, Whitley R, Yamanishi K, editors. Human

Herpesviruses: Biology, Therapy, and Immunoprophylaxis. Cambridge: Cambridge

University Press; 2007. Chapter 69. PMID: 21348132.

Ramachandran S, Kinchington PR. Potential prophylactic and therapeutic

vaccines for HSV infections. Curr Pharm Des. 2007;13(19):1965-73. doi:

10.2174/138161207781039760. PMID: 17627530.

Us D. Dünden bugüne herpes simpleks virus aÅŸi çaliÅŸmalari [Herpes simplex

virus vaccine studies: from past to present]. Mikrobiyol Bul. 2006

Oct;40(4):413-33. Turkish. PMID: 17205702.

Koelle DM. Vaccines for herpes simplex virus infections. Curr Opin Investig

Drugs. 2006 Feb;7(2):136-41. PMID: 16499283.

Rajcáni J, Durmanová V. Developments in herpes simplex virus vaccines: old

problems and new challenges. Folia Microbiol (Praha). 2006;51(2):67-85. doi:

10.1007/BF02932160. PMID: 16821715.

Bernstein D. Glycoprotein D adjuvant herpes simplex virus vaccine. Expert

Rev Vaccines. 2005 Oct;4(5):615-27. doi: 10.1586/14760584.4.5.615. PMID:

16221064.

Rajcáni J, Mosko T, Rezuchová I. Current developments in viral DNA vaccines:

shall they solve the unsolved? Rev Med Virol. 2005 Sep-Oct;15(5):303-25. doi:

10.1002/rmv.467. PMID: 15906276.

Jones CA, Cunningham AL. Vaccination strategies to prevent genital herpes

and neonatal herpes simplex virus (HSV) disease. Herpes. 2004 Apr;11(1):12-7.

PMID: 15115632.

Comments on the Quality of English Language

Minor editing will help but it is not an issue

Reviewer 3 Report

Comments and Suggestions for Authors

The manuscript by Ren et al. describes the construction and in-vivo testing of an HSV-2 therapeutic vaccine. Their approach is based on the expression of HSV-2 gD and ICP27 in the Sendai virus replication-defective vectors. The resulting recombinant viruses were tested in mice to optimize dosage and regimen and in guinea pigs to assess their therapeutic efficacy. Overall the data suggest that the treatment regimen resulted in some reduction of severity of recurrent lesions, and duration and frequency of recurrences as compared to negative control (PBS). The study is not really novel (both gD and ICP27 were previously explored as components of subunit vaccines), however, it provides an incremental improvement in understanding of the topic. The results seem to be straightforward, but their presentation and description lack clarity, logic, and purpose. The specific comments are listed below:

Line 99 - PFU is plaque-forming units (not particle forming units)

Line 101 - indicate the strain of the mice used (I know it is indicated in other places, but this is the correct place to describe it).

Line 112 - what volume was injected i.m.?

Table 1 is very chaotic and confusing, different treatment groups need to be spelled out and properly described

Results 3.1 - The justification for using 10^8 CIU doesn’t make much sense. The 10-fold difference in dose will not compensate for the ~2000-fold difference in body weight between mice and humans but may be quite significant from the safety standpoint. 

Line 383 - for the simultaneous vaccination, did the authors use 10^8 CIU of each virus or combined (that is 10^4 of each virus)?

Results 3.2 - the study is missing an important control - SeV -F alone, how do the authors measure the contribution of the vector to the observed effects?

Results 3.2 - why no immunology work was performed in guinea pigs? How do the authors know that the observed effects are the result of HSV antigen expression and not something else? With this experimental design, the correlative effects are not established. 

The discussion reads as an introduction and doesn’t really “discuss” the data. The discussion needs to be re-written to discuss the data from this study in the context of knowledge available in the field.

For all figures - the figure legends poorly describe the context of the figure leaving readers guessing what they are looking at.

Comments on the Quality of English Language

Language: multiple typos, inconsistencies, and grammatical errors throughout the whole manuscript. While I realize that the authors are not native English speakers and appreciate their effort, the manuscript needs to be reviewed/edited by a professional editor to improve clarity and make it more readable.

Reviewer 4 Report

Comments and Suggestions for Authors

The manuscript describes the evaluation of Sendai virus-based recombinants expressing HSV-2 gD and HSV-2 ICP27 as the basis for a protective vaccine against genital HSV-2 reactivated infection.  The immunological studies in the BALB/c mouse model are sound and well presented, while the protective studies in the guinea pig model are suggestive of efficacy.  However, there are some issues with the study that diminish overall enthusiasm.

The authors state that HSV-2 gD and HSV-2 ICP27 were chosen to induce humoral and cellular immunity, respectively, in BALB/c mice.  However, there is no explicit explanation why these viral proteins were chosen.  While it is well known that gD induces neutralizing antibodies, it should be stated what role ICP27 plays in inducing CD4 or CD8 T cell responses.

The authors present strong evidence of an effective HSV-2-specific antibody response and T cell response following immunization with the recombinant gD- and ICP27-expressing Sendai virus vectors.  The analysis of the neutralizing antibody titers and the cytokine profiles induced by both CD4 and CD8 T cell subpopulations indicate the validity of the immunization protocol in BALB/c mice in the induction of an immune response.  However, it is not demonstrated, in the mouse system, that such an immune response is protective.  This would have strengthened the findings of the manuscript.

For the measurement of protection, the authors switch to a guinea pig model.  The choice of the guinea pig model, in which latent HSV-2 reactivates and induces secondary lesions, is valid to the study.  However, the authors have not shown that the recombinant Sendai virus vectors induce the same immune response in the guinea pig model.  While it is clear that both gD and ICP27 induce B and T cell responses, respectively, in BALB/c mice, it is not clear that the same is true for guinea pigs.  While it is likely that gD will serve as a target for B cells and the induction of neutralizing antibodies, it is not clear that ICP27 contains epitopes that can bind to guinea pig MHC and induce an appropriate T cell response.  It would have strengthened the study to include a group of animals that received only the gD vaccine alone to determine if the ICP27-expressing recombinant played any role in the guinea pig model.

 Overall, there are some strong aspects to the study, but there is a disconnect between the immunogenic studies in the mouse and the protective studies in the guinea pig.  Reconciling these issues would strengthen the manuscript.

Comments on the Quality of English Language

The quality of the language is excellent.  There are a few misspellings to be addressed in editing.

Round 2

Reviewer 1 Report

Comments and Suggestions for Authors

The authors have not adequately substantiated that the observed protective effects in the guinea pig model are a direct result of the vaccine-induced immunological responses, as seen in the mouse model. To enhance the robustness of their findings, it is imperative that the authors consider demonstrating the vaccine's efficacy in a mouse model of HSV-2 infection. This step would provide a critical link between the vaccine's immunological responses and its protective effects, thereby strengthening the overall scientific rigor of their investigation

Comments on the Quality of English Language

Minor editing will help but it is not an issue.

Reviewer 2 Report

Comments and Suggestions for Authors

I had provided a list of most relevant papers that must be cited and their findings compared with the conclusions drawn in this research; none of the references I suggested were included in the revision; please explain why is the this inclusion not acceptable.

Comments on the Quality of English Language

Substantial improvement required;

Round 3

Reviewer 1 Report

Comments and Suggestions for Authors

The authors have addressed my concerns.

Reviewer 2 Report

Comments and Suggestions for Authors

The authors have added relevant citations, even though more are needed but sufficient. 

Comments on the Quality of English Language

Also, the writing style is more informal and not suitable for research publications. Authors can benefit from the editing services of MDPI or another service.